# Balance Ability Characteristics and Related Factors in Athletes Across Different Sports: A Preliminary Study

**DOI:** 10.3390/healthcare12222240

**Published:** 2024-11-11

**Authors:** Yasuhiro Suzuki, Yukiyo Shimizu, Kazushi Maruo, Takumi Tsubaki, Yuuki Tanabe, Yasushi Hada

**Affiliations:** 1Department of Rehabilitation Medicine, University of Tsukuba Hospital, Tsukuba 305-8576, Ibaraki, Japan; tsubaki2308@gmail.com (T.T.); yuuki0712rina@yahoo.co.jp (Y.T.); 2Institute of Systems and Information Engineering, University of Tsukuba, Tsukuba 305-8573, Ibaraki, Japan; 3Biomedical Science and Engineering Research Center, Hakodate Medical Association Nursing and Re-Habilitation Academy, Hakodate 040-0081, Hokkaido, Japan; 4Department of Internal Medicine (Endocrinology and Metabolism), Institute of Medicine, University of Tsukuba, Tsukuba 305-8576, Ibaraki, Japan; 5Department of Rehabilitation Medicine, Institute of Medicine, University of Tsukuba, Tsukuba 305-8575, Ibaraki, Japan; shimiyukig@md.tsukuba.ac.jp (Y.S.); y-hada@md.tsukuba.ac.jp (Y.H.); 6Department of Biostatistics, Institute of Medicine, University of Tsukuba, Tsukuba 305-8575, Ibaraki, Japan; kazushi.maruo@gmail.com

**Keywords:** modified index of postural stability, one-leg standing, stabilometry

## Abstract

Background: We conducted a cross-sectional study to examine two-leg- and one-leg-type balance characteristics in athletes and explore factors related to their balance ability. Methods: A total of 213 participants, including athletes from various sports (gymnastics, boat racing, swimming, soccer, judo, and baseball) and non-athletes, were included (142 men, 71 women, average age 21.5 ± 2.1 years). Balance ability was classified into two-leg and one-leg types using the modified index of postural stability (mIPS) in a two-leg stance and the one-legged stance duration with eyes closed (OLS). Body composition, upper and lower limb strength, and lower limb sensation were also measured. To examine the balance characteristics of each sport, the mIPS and OLS were used as dependent variables in a multiple regression model with age, height, weight, and sex as independent variables. Results: The results showed a significantly higher mIPS in gymnastics (estimate: 0.22) and boat racing (0.14), and it was lower in swimming (−0.25). The OLS was significantly higher in soccer (16.98), judo (16.23), gymnastics (9.77), and baseball (9.12) and significantly lower in swimming (7.93). Additionally, the mIPS was independently associated with knee extension strength (0.12), sensory motor variables (−0.004), and height (−0.01). The OLS was associated with skeletal muscle mass (1.85) and height (−1.42). Conclusions: In summary, gymnasts showed superior two-leg and one-leg balance; boat racers excelled in two-leg balance; swimmers showed inferior two-leg but better one-leg balance; and soccer, judo, and baseball athletes demonstrated superior one-leg balance. Additionally, the mIPS was associated with knee extensor strength, plantar pressure sensation, and height, whereas the OLS was associated with skeletal muscle mass and height.

## 1. Introduction

Balance is an indispensable motor skill based on muscular synergies that minimize the displacement of the center of pressure while maintaining an upright stance, proper orientation, and adequate locomotion. Balance ability is classified into static and dynamic categories [1,2]. Maintaining static or dynamic balance is essential for good performance in various sports [2]. Athletes generally require training to improve their body balance, which is crucial for competitive performance [3]. Balance training can not only improve an athlete’s functional performance, posture, and neuromuscular control, but also reduce the risk of sports-related injuries [4,5,6]. Meta-analyses on the effects of balance training in youth have reported moderate-to-large effects on both static and dynamic balance [7], highlighting the importance of incorporating balance training in the training programs of young athletes [8]. In a review, Hrysomallis compared the balance abilities of athletes across different sports horizontally. It was found that gymnasts possessed superior balance abilities, followed by soccer players, swimmers, controls, and basketball players [9]. Additionally, a report comparing horseback riders, judokas, and non-athletes found that horseback riders had the best balance abilities [10]. Knowing the balance characteristics of each competition would assist in devising more effective balance exercises. Therefore, it is vital to recognize performance variations when prescribing balance exercises to athletes in different sports.

Researchers have used various balance tests to assess balance when investigating the relationship between balance ability and athletic performance. Hrysomallis discussed the balance abilities of athletes for each leg, considering both two-leg and one-leg standing conditions [9]. The frequency of two-leg and one-leg standing varies considerably between sports and may reflect specific characteristics of each sport. For example, elite golfers demonstrated better one-leg balance than less skilled players [11]. Golfers also need to swing in unstable conditions, such as on slopes or with one foot on sand and the other on grass, which is considered one reason why one-legged balance is important [11]. One approach to clarify athletes’ balance characteristics is to classify each sport as either two-leg or one-leg types.

Regarding two-leg-type balance characteristics, we focus on the modified index of postural stability (mIPS) [12]. Our previous research has shown that the mIPS is associated with competitive performance in athletes, specifically boat racers [13,14]. Therefore, the mIPS can be considered an appropriate index for assessing two-leg balance characteristics. In addition, conducting quantitative assessments of balance ability using continuous variables allowed us to explore the associated factors. In a review, balance components were identified as including the underlying motor systems (100% of measures), anticipatory postural control (72%), static stability (62%), sensory integration (52%), dynamic stability (48%), functional stability limits (24%), cognitive influences (24%), verticality (9%), and reactive postural control (0%) [15]. According to Davlin, athletes’ dynamic balance abilities are associated with height and weight [3], suggesting the potential influence of physique, making it meaningful to focus on body composition factors such as skeletal muscle mass and body fat mass. When focusing on realistically measurable items in our clinic, factors such as the musculoskeletal system measured through strength assessments, sensory integration measured through sensory assessments, and body composition influencing physique were identified as measurable items. However, no studies have comprehensively examined the relationship between these factors and balance.

This study aimed to examine the balance characteristics (two-leg and one-leg types) for each competition using a unified quantitative evaluation index of balance ability index, including the mIPS. Additionally, we investigated physical functions associated with balance ability (two-leg and one-leg types), focusing on factors like somatosensory function, muscle strength, and body composition. As this is a preliminary study, the sports were not predetermined and participants were grouped based on recruitment. However, due to the established association of the mIPS with boat racing, it was included as one of the target sports.

## 2. Materials and Methods

The target participants were university students and boat race trainees. They were recruited through the bulletin boards of regional universities and boat racing training schools between February 2019 and February 2021. All participants were university sports staff, non-staff, and boat race trainees. The control group comprised healthy university students (non-athletes) who did not belong to a university sports club. Participants with impaired balance ability due to any disorder or disability were excluded based on the following conditions: (i) spontaneous nystagmus; (ii) visual impairment or limb movement disorders affecting daily living; (iii) inability to maintain normal standing posture; (ⅳ) dizziness or vertigo; (v) history of equilibrium sensory disorder; (ⅵ) history of falls; and (ⅶ) lack of independence in walking and daily living. The purpose of the research and the risks of the measurements were explained to each participant, and written informed consent was obtained from all participants. This study was approved by the Ethics Committee of the University of Tsukuba Hospital (protocol number H26–29) and was conducted according to the tenets of the Declaration of Helsinki. Additionally, this study adhered to ethical standards throughout the research process.

### 2.1. Measurements

#### 2.1.1. Selection of Measurement Items

For the two-leg-type balance characteristic, the modified index of postural stability (mIPS) measured during two-leg standing was chosen. The mIPS serves as a quantitative indicator with reported normative values across a wide range of age groups [12].The one-leg standing test, a simple field test involving the measurement of the duration of one-leg standing, was selected to evaluate the one-leg balance characteristic [11,16]. The test was conducted using the one-leg standing time with eyes closed (OLS) to minimize the ceiling effect [17]. The advantage of the OLS is that it can evaluate differences in competitive levels and can reflect performance in specific sports, such as in relation to the level of proficiency in soccer [18,19]. In this study, the factors related to balance ability were narrowed down to basic attributes (age, sex, height, and weight), muscle strength, sensory function, and body composition. Muscle strength was assessed through measurements of knee extension strength, knee extension muscle endurance, ankle dorsiflexion muscle strength, toe pinch force, and grip strength. Sensory function was assessed through measurements of vibration sensation (deep sensation) and plantar touch sensation (superficial sensation). Body composition measurements included skeletal muscle mass and body fat mass. All measurements were performed in accordance with methods used in previous studies [20,21] and were conducted consistently across all subjects.

#### 2.1.2. Measurement of Balance Ability

The mIPS was measured using a gravicorder (GP-6000, Anima Co., Tokyo, Japan). Participants were measured while standing with their eyes closed on a foam rubber (Airex Balance-pad Elite, Airex AG, Sins, Switzerland) placed on the plate of the gravicorder. First, the participant stood in a resting position with the inside of the foot at a distance of 10 cm on a Gravicorder. This was performed to measure instantaneous fluctuations in the center of pressure (COP) at a sampling frequency of 20 Hz. The participant was then instructed to incline their body to the front, rear, right, and left while keeping their body straight and without moving their feet. Instantaneous fluctuations in the COP were measured at each position. The mIPS was calculated as follows: log[(area of stability limit + area of postural sway)/area of postural sway]. The stability limit area was calculated as follows: (front and rear center movement distance between the anterior and posterior positions) × (distance between the right and left positions). The area of postural sway was calculated as the average measurement value in 10 s under the anterior, posterior, right, left, and center positions. The area of postural sway was calculated as the mean sway area of the five positions [12].

The OLS was based on the method of Bohannon et al. [22], in which participants performed one-leg standing with their eyes closed. Participants were required to maintain one-leg standing barefoot with their arms held by their sides and eyes closed. The goal of each activity was to maintain balance for 60 s. Two attempts were permitted for each participant, and the test was performed with each foot. If the participant reached this goal, 60 s was recorded. If the participant did not reach the goal, the best of the two trials was recorded. There was a rest period of 1 min between trials and a rest period of up to 5 min between trials was granted at the participant’s request.

#### 2.1.3. Knee Extension Strength and Knee Extension Muscle Endurance

Knee extension muscle strength and knee extension muscle endurance were measured on the dominant foot side using a torque machine (Biodex System3; Sakai Medical, Tokyo, Japan). Knee extension muscle strength was determined by performing three consecutive knee extension operations with maximum effort in isokinetic muscle strength measurements (60°/s). The maximum torque value (Nm/kg) was used as the representative measurement value. Knee extensor muscle endurance was determined by measuring the total work (J) of continuous knee extensions 20 times with maximum effort in isokinetic muscle strength measurements (300°/s).

#### 2.1.4. Dorsiflexion Strength of the Ankle Joint

The dorsiflexion strength of the ankle joint was measured using a handheld dynamometer (Anima Co., Tokyo, Japan). While seated, the participants adjusted the height of the chair such that the angle between the knee and ankle joints was 90°. The participant then placed the heel on the floor and dorsiflexed the foot to the maximum. The examiner attached the handheld dynamometer to the back of the participant’s foot. Maximum pressure was applied in the ankle plantar flexion direction to break the participant’s maximum ankle dorsiflexion. The average values (kgf) of the left and right sides of the isometric dorsiflexor muscle strength of the ankle joint were used. The test was performed twice on both the left and right sides and the best results for the left and right feet were averaged.

#### 2.1.5. Toe Pinch Force

Toe pinch force was measured using a pinch force dynamometer (Checker-kun, Nisshin Sangyo Inc., Saitama, Japan). The participants sat on a chair with their arms crossed over their chests. The dynamometer was attached to the foot of the participant between the big toe and second toe. This was performed while the participant was in a sitting position, with the hip and knee joints at 90° of flexion. The test was performed twice for each foot and the best results for the left and right feet were averaged [23].

#### 2.1.6. Grip Strength

The grip strengths of the right and left hands were measured using a Smedley analog grip meter (ST100 T-1780; Toei Light Co., Tokyo, Japan). The maximum value (kgf) was considered as the measured value.

### 2.2. Sensory Tests

#### 2.2.1. Deep Sensation

Vibration perception testing for deep sensation was performed by applying a vibrating tuning fork (C128 Hz aluminum tuning fork; Niti-on, Chiba, Japan) to the medial malleolus, with the participants lying in the supine position. The participants were asked to raise their hands to indicate when they no longer felt the vibration. Testing was conducted twice on each leg, and the average value of the perception duration was taken as the representative value.

#### 2.2.2. Superficial Sensation

Semmes–Weinstein monofilament testing (SWMT) for superficial sensation was performed with a set of 20 monofilaments (SOTDM20A; Sakai Medical, Tokyo, Japan). Monofilaments were randomly applied to the calcaneal region of each leg in ascending order of the target force while avoiding heavily calloused areas. The participants were blindfolded, and the monofilaments were pressed against the skin at each location until they were bent and held in place for 1.5 s. The participants were asked to describe the location on the foot where they sensed the monofilament. The SWMT threshold was defined as the pressure corresponding to the minimum target force sensed by the participants in the three trials.

### 2.3. Body Composition

Body composition was assessed by measuring skeletal muscle mass and body fat mass and was measured using bioelectrical impedance analysis (InBody 720; Biospace, Tokyo, Japan).

### 2.4. Statistical Analysis

The descriptions of continuous variables (age, height, weight, body mass index) for all characteristics of participants are presented as the mean ± standard deviation (SD). Continuous variables were assessed for normal distribution using the Shapiro–Wilk test and are expressed as the mean ± SD or median (25 percentile, 75 percentile) based on distribution. A competitive event in which ten or more participants gathered was certified as a sporting group. To investigate the balance characteristics of each sport, we applied a multiple regression model. The model included the balance ability index (mIPS or OLS) as the dependent variable, and each sports group variable, age, height, weight, and sex as independent variables.

Subsequently, participants without missing data for each factor were selected. Muscle strength, sensory, and body composition factors related to the balance ability index (mIPS and OLS) were then examined. We applied a multiple regression model, including the mIPS or OLS as the dependent variable and all factors (basic attributes, muscle strength, sensory function, and body composition) as independent variables.

Variable selection for each regression model was conducted using a stepwise procedure based on the Akaike information criteria. All statistical analyses were performed using R software ver. 4.3.2 (R Core Team, Vienna, Austria), and the criterion for statistical significance was set at *p* < 0.05.

## 3. Results

A total of 213 participants were included, divided into 11 groups, namely control, judo, swimming, soccer, baseball, gymnastics, American football, equestrian sports, lacrosse, tennis, and boat race trainees. Seven groups (control, judo, swimming, soccer, baseball, gymnastics, and boat race trainees) had more than 10 participants. These groups included 176 athlete participants (150 university students plus 26 boat race trainees; 107 men, 69 women; age, 20.2 ± 1.6 years; height, 1.69 ± 0.08 m; and weight, 68.0 ± 14.7 kg) and 37 non-athletes as control participants (university students) (Table 1). All university athletes had undergone technical training for at least 6 years during junior high school and high school. Non-athletes had never performed such technical training in any sport. None of the participants had injuries that inhibited maximal exertion or conditions that were likely to be aggravated by maximal exertion. Due to the occurrence of a major infectious disease outbreak (pandemic) during the participant recruitment period, measures were taken to shorten the assessment time of the evaluation items as an infection control measure. The impact of the pandemic led to two main issues, namely (i) participants requested a shorter assessment time and (ii) the research team had to adjust the environment (limited the number of staff) to conduct the assessments. This resulted in an imbalance in the number of assessments for each sport. As a result, many participants experienced missing data in the assessment items; ultimately, only 113 participants completed all the assessments. The following are the number of missing data points for each assessment item: knee extension strength (91), knee extension endurance (91), ankle joint dorsiflexion strength (97), toe pinch force (60), grip strength (90), SWMT (69), vibration perception test (60), and body composition (98).

The results of the multiple regression analysis are shown below. Explanatory variables that were not selected are considered to have had no significant effect on the outcome. The characteristics of the participants, the mIPS, and the OLS are presented in Table 1. Table 2 and Table 3 present the results of balance tests (mIPS and OLS) for each sports group (judo, swimming, soccer, baseball, boat race training, and gymnastics) compared to the control group. The mIPS was significantly higher in gymnastics (*p* < 0.001) and boat race athletes (*p* = 0.005) and significantly lower in swimming athletes (*p* < 0.001). The OLS was significantly higher in soccer (*p* < 0.001), judo (*p* < 0.001), gymnastics (*p* = 0.005), and baseball (*p* = 0.014) and swimming (*p* = 0.021). In summary, not all sports showed significantly higher mIPS and OLS scores than those of the control group.

Table 4 shows the attributes of the participants who performed all body composition, muscle strength, balance ability, and sensory tests. Among these participants, swimmers and soccer players accounted for more than 60%. Table 5 and Table 6 present the results of the stepwise analysis, including all variables as independent variables and using the mIPS and OLS as dependent variables. The mIPS was independently associated with knee extension strength (*p* = 0.006), sensory motor variables (*p* = 0.027), and height (*p* = 0.018). In summary, the mIPS increased with stronger knee extensions, more sensitive plantar pressure sensation, and shorter heights. The OLS was independently associated with skeletal muscle mass (*p* = 0.021) and height (*p* < 0.001). In summary, the OLS increased with greater skeletal muscle mass and shorter heights.

## 4. Discussion

In this study, we targeted athletes in each sport and classified the balance characteristics of each sport as two-leg or one-leg using the mIPS and OLS. As a result, gymnastics was identified as an excellent two-leg and one-leg balance type; boat racing was identified as an excellent two-leg type; swimming was a subpar two-leg type and an excellent one-leg type; and soccer, judo, gymnastics, and baseball were excellent one-leg types. Using this classification, it was possible to categorize each sports discipline into four types of balance characteristics.

In this study, the mIPS and one-leg standing time with eyes closed (OLS) were utilized as indicators of balance ability, each reflecting the ability to maintain balance in a standing position with eyes closed. This is considered an indicator that allows for a more sensitive assessment of balance ability, which is dependent on vision. In a previous study of soccer players, professional players had significantly less center-of-gravity sway during closed-eye standing than amateur players, and the results were almost the same in the open-eyes standing position [24]. In other words, players with more advanced soccer skills could control their center of gravity without relying on visual information and without supervision [24]. In addition, in a group of 13-year-old boys, athletes with soccer experience showed less visual dependence on balance ability than non-athletes based on the results of the center-of-gravity sway test [25]. Subsequently, an improvement in soccer skills should increase control of the center of gravity (balance ability) using non-visual information without relying on visual information. Similarly, athletes in all competitions in which the mIPS or OLS scores are high may have competitive characteristics that influence balance ability without relying on visual information.

### 4.1. Gymnastics

Gymnasts often practice two-leg stance balance skills on balance beams, similar to the skills required in the mIPS. Hence, gymnasts may develop a superior attention focus on cues that alter the balance performance unique to two-leg standing, such as small changes in joint position and acceleration. Gymnasts have many opportunities for one-leg stance positions during performances, such as floor exercises, and it has been suggested that they may possess equal or superior one-leg balance abilities compared to soccer players [26]. Indeed, they have been reported to excel in one-leg stance ability compared to athletes in other sports and control groups [3,27,28]. From these findings, it can be inferred that the gymnasts in this study possessed excellent balance abilities on two legs and one leg.

### 4.2. Boat Racing

In boat racing, the operator’s center of gravity shift on curves is crucial for steering the boat on a lap course, which requires advanced dynamic balance capabilities with two-leg standing. The mIPS is an indicator of dynamic balance ability that assesses the ability to shift the center of gravity in the anterior–posterior and left–right directions during two-leg standing [12,14]. Therefore, this may reflect the boat operators’ steering abilities during training. Furthermore, despite the boat trainees being in a crouched position, the participants exhibited high balance ability in two-leg standing. In previous studies, equestrian athletes were reported to have less sway in the center of gravity in two-leg standing with closed eyes and rubber than non-athletes [29]. This suggests that even in sports predominantly performed in a seated position, control of the center of gravity during two-leg standing is well managed. This result was also reflected in the balance indicated by the mIPS, showing outcomes similar to those in the present study.

### 4.3. Swimming

According to Ide et al., since the mIPS is performed under eyes-closed and foam rubber standing conditions, it has been reported to be an index reflecting vestibular balance function [30,31]. This is inferred from the characteristics of the measurement, where in an environment that hinders vision (closed eyes) and proprioception (standing on foam rubber), the load on the vestibular labyrinth system increases when the head is voluntarily moved forward, backward, left, and right in a two-leg stance. This heightened load makes it easier to disrupt balance. However, in the case of the OLS with eyes closed, the patient stands on one leg on a hard surface, where the support base is small. Additionally, because the measurement protocol required minimal head movement, the load on the vestibular labyrinth system was reduced. Swimmers typically do not engage in static or dynamic balance exercises on one or two legs and potentially lack significant stimuli to enhance their balance abilities [15]. Essentially, swimmers may experience reduced stimulation of the vestibular labyrinth system because of prolonged and extended activities in a non-weight-bearing environment, potentially leading to diminished function.

However, regarding the superior performance of swimmers in the one-leg stance (OLS), it is possible that the results reflected the core muscle strength of swimmers. Improvements in trunk stability owing to core muscle activity have been shown to enhance swimming performance [32]. Moreover, balance in the one-leg stance (eyes open) has been reported to be more positively correlated with trunk muscle activity than two-legged balance [33]. In other words, swimmers are considered to have improved stability in the OLS because of the stability of their cores. Based on these findings, it can be inferred that the swimmers in this study had inferior two-legged and superior one-legged balance.

### 4.4. Soccer

Soccer players frequently perform dynamic movements on one leg when kicking the ball [3]. Therefore, soccer players are expected to excel more in the one-legged stance than athletes in other sports [26]. The results of this study suggest that soccer players make greater use of the somatosensory system during one-leg standing than other athletes, supporting the results of previous studies.

### 4.5. Judo

Characteristically, judo athletes in competition primarily focus on techniques aimed at disrupting the opponent’s balance and causing them to fall, requiring effective control of dynamic postures. Therefore, they frequently use the one-leg stance to apply techniques to their opponents during matches and practice. In other words, considering the competitive nature of judo, the judo group in this study had excellent one-leg balance abilities.

### 4.6. Baseball

The one-leg stance of baseball pitchers has been reported to have no correlation with pitching accuracy [34]. However, in the context of the relationship between baseball hitting and the one-leg stance, the action of supporting body weight in a one-leg stance during batting may potentially affect the stability and hitting power of the batter. During batting, the batter places weight on one foot while lifting the other. The more stable the one-leg stance position, the easier it is for the batter to maintain swing control. Good stability in the one-leg stance enables accurate swinging against the ball, contributing to an increased likelihood of successful hits. Thus, baseball players can enhance their one-leg stance abilities through repeated batting.

In summary, as a competitive characteristic of baseball, the baseball group in this study showed potential excellence in one-legged balance.

### 4.7. Summary

To devise effective balance exercises to improve athletic performance, it is important to focus on the indicators of balance ability and explore the related factors. This could also lead to injury prevention for athletes. After analyzing which factors related to balance ability, we examined the relationships between balance ability, body composition, and muscle strength. The participants in this factor analysis were young individuals, including athletes, and the results provide fundamental information for understanding the constituent factors of balance ability.

As a result, it became evident that the mIPS increased with a shorter height, higher knee extension muscle strength, and more acute plantar pressure sensitivity. Additionally, the OLS increased with a shorter height and higher skeletal muscle mass. Davlin [3] reported that an athlete’s balance ability (continuous postural retention in an unstable horizontal position) was negatively correlated with height. Similar results were obtained in the present study. The factors that affect balance ability include a low position of the sacral region, where the center of gravity of the body is located, and a small physique. These factors may favor balance ability. For example, when analyzing comparative studies, it should be noted that gymnasts tend to be shorter than other athletes and that height may influence balance ability [3]. The normalization of balance scores relative to height or limb length should be considered when comparing groups with notable differences in stature or weight [35].

Previous studies reported that plantar mechanoreceptors provide information to the central nervous system regarding the vertical orientation of the body [36]. This information from the plantar surface is believed to involve tactile sensitivity, and the postural adjustment function of the plantar surface suggests that athletes demonstrate superior performance in closed-eye environments. This also indicates the importance of proprioception in maintaining a stable upright posture [37]. The mIPS is a balance test performed in closed-eye and standing conditions on a rubber surface and has been suggested as an indicator of tactile sensitivity from the plantar surface.

Previous studies reported a positive correlation between balance ability and knee extension muscle strength in balance tests using perturbations to tilt the upright posture forward and backward [38]. This balance test from previous studies involved applying external perturbations. During the maintenance of posture under these perturbations, muscle activity propagated from the distal to the proximal muscles [39]. Therefore, it is believed to be related to knee extension muscle strength. The mIPS, which is a balance test similar to that used in previous studies, may also involve knee extension muscle strength through a similar mechanism.

Balance in a one-leg stance has been reported to be positively correlated with trunk muscle activity [33]. However, trunk muscle mass may also be associated with this correlation. As trunk muscle mass constitutes more than half of the total skeletal muscle mass in the body, there is a potential association between the one-leg stance and skeletal muscle mass.

In this study, knee extension strength, proprioception, height, and skeletal muscle mass were identified as factors related to balance ability. These factors may contribute to improvements in balance ability. Therefore, we propose training programs to enhance balance ability by utilizing this knowledge. The first approach is strength training. If strength training increases knee extension strength and muscle hypertrophy, it may improve the stability of the lower limb joints and neuromuscular coordination, indirectly contributing to enhanced balance ability. However, limited evidence on the direct effect of strength training on improving balance ability itself exists [9]. In other words, while muscle strength and muscle mass are necessary conditions for balance ability, strength training alone may have a limited impact. The second method is proprioception training. It has been reported that proprioception training for female volleyball players significantly improved their balance ability and competitive performance [40]. This suggests that proprioception has a causal relationship with balance ability and may be an effective training method for athletes. The third focus is balance training. This study revealed that balance types (two-leg and/or one-leg) vary by sport, making it important to design training programs specific to each type. For instance, it has been reported that balance training focused on one-leg training, tailored to the characteristics of the sport, may improve balance ability and technical skills in male soccer players [41]. If the balance characteristics determined in this study reflect the characteristics of the sport, the mIPS and OLS are useful indicators for designing balance training. Finally, it is essential to consider an athlete’s height when designing training programs. Setting the difficulty level based on age standards for height can be beneficial, as height can influence changes in balance ability.

### 4.8. Limitations

This study has several limitations. First, the composition of sports groups may be biased in the examination of factors related to dynamic balance ability; thus, the conclusions obtained in this study cannot be applied to all athletes. Additionally, it was not possible to perform an analysis for each sport nor to clarify the relationships between muscle strength, proprioception, and body composition, which are balance components specific to each sport. In future analyses, adjusting the composition of sports groups should be considered.

Second, there was an imbalance in the number of participants across sports groups. Sports with fewer than 10 participants were not included in the model analysis, so effects for these sports were not evaluated. Furthermore, because this was an exploratory study, the sample size was not designed based on statistical power, raising concerns that certain effects may not have been detected due to insufficient power. Increasing the sample size is recommended in future studies.

Third, the competitive level of participating athletes in each sport was not clearly defined, so it could not be standardized. In some sports, differences in balance ability have been shown to depend on participant skill level, even within the same discipline [42,43,44]. Thus, both the type of sport and the proficiency level of athletes can influence balance ability. Data on athletic proficiency and skill levels are necessary to interpret this study’s result accurately, and standardizing skill levels across sports would be beneficial.

Fourth, the gender distribution in each sports group was unbalanced. Females may have lower physical strength and athletic performance than males, potentially affecting the results. Since this was an exploratory study, adjustments were made for female participants; however, for conclusive findings, the gender of participants should be standardized.

Fifth, the coefficient of determination in the multiple regression analysis was not high enough. Therefore, the accuracy of the analysis is limited, and further investigation based on this study’ finding is required.

## 5. Conclusions

In this study, we targeted athletes from various sports and classified their balance characteristics of each sport as either two-leg or one-leg balance types using the mIPS and OLS. Gymnastics was identified as having an excellent two-leg and one-leg balance; boat racing showed excellent two-leg balance; swimmers demonstrated inferior two-leg but better one-leg balance; and soccer, judo, and baseball showed excellent one-leg balance. An analysis of the factors related to balance ability revealed that the mIPS increased with a shorter height, greater knee extension strength, and higher acute plantar pressure sensitivity. In contrast, the OLS was associated with a shorter height and greater skeletal muscle mass.

## Figures and Tables

**Table 1 healthcare-12-02240-t001:** Characteristics of the participants in each sports group.

Sporting Group	*n*	Female Sex, *n* (%)	Age (Year)	Height (m)	Weight (kg)	Body Mass Index (kg/m^2^)	mIPS	OLS (s)
All	213	71 (33.3)	21.5 ± 2.1	1.68 ± 0.08	64.2 ± 14.5	22.6 ± 3.9	0.73 ± 0.25	57 (37, 60)
Control	37	20 (54.1)	21.5 ± 2.0	1.64 ± 0.08	58.1 ± 10.6	21.3 ± 2.5	0.73 ± 0.22	42 (26, 60)
Judo	34	11 (32.4)	20.2 ± 1.3	1.70 ± 0.08	78.1 ± 22.1	26.9 ± 6.1	0.71 ± 0.22	60 (38, 60)
Swimming	33	14 (42.4)	19.9 ± 1.4	1.71 ± 0.08	65.4 ± 8.0	22.4 ± 1.4	0.50 ± 0.21	53 (21, 60)
Soccer	30	10 (33.3)	19.8 ± 1.2	1.67 ± 0.07	62.4 ± 8.6	22.2 ± 2.0	0.81 ± 0.20	60 (60, 60)
Baseball	27	0 (0)	19.6 ± 1.0	1.74 ± 0.05	71.2 ± 12.7	23.5 ± 4.2	0.66 ± 0.23	54 (34, 60)
Boat race trainee	26	8 (30.8)	21.8 ± 3.4	1.63 ± 0.07	51.3 ± 3.2	19.4 ± 1.0	0.92 ± 0.21	56 (36, 60)
Gymnastics	15	6 (40.0)	22.3 ± 2.5	1.63 ± 0.08	57.5 ± 7.2	21.5 ± 1.2	0.96 ± 0.24	57 (37, 60)
American football	3	0 (0)	20.7 ± 2.1	1.71 ± 0.02	73.2 ± 5.3	25.1 ± 1.2	0.60 ± 0.28	45 (41. 52)
Equestrian sports	3	0 (0)	20.7 ± 0.6	1.72 ± 0.07	60.7 ± 1.5	20.7 ± 1.7	0.78 ± 0.34	60 (42, 60)
Lacrosse	3	0 (0)	20.0 ± 0.0	1.73 ± 0.03	70.2 ± 1.8	23.4 ± 0.3	0.74 ± 0.17	39 (30, 45)
Tennis	2	2 (100)	19.0 ± 0.0	1.64 ± 0.04	56.5 ± 2.0	21.8 ± 2.0	0.83 ± 0.12	43 (42, 43)

Data are in the form of mean ± standard deviation or median (25th, 75th). mIPS, modified index of postural stability; OLS, one-leg standing time with eyes closed.

**Table 2 healthcare-12-02240-t002:** Multiple regression analysis of the mIPS.

R^2^ = 0.30	Estimate	95% Confidence Interval	*p* Value
Gymnastics	0.22	0.10–0.34	<0.001
Boat race trainee	0.14	0.04–0.23	0.005
Swimming	−0.25	−0.33–−0.17	<0.001
Baseball	−0.08	−0.17–0.02	0.102

Covariates: age, height, weight, and sex; mIPS, modified index of postural stability.

**Table 3 healthcare-12-02240-t003:** Multiple regression analysis using the OLS.

R^2^ = 0.23	Estimate	95% Confidence Interval	*p* Value
Soccer	16.98	10.46–23.49	<0.001
Judo	16.23	8.97–23.49	<0.001
Gymnastics	9.77	1.31–18.24	0.024
Baseball	9.12	1.89–16.36	0.014
Swimming	7.93	1.22–14.63	0.021

Covariates: age, height, weight, and sex; OLS, one-leg standing time with eyes closed.

**Table 4 healthcare-12-02240-t004:** Characteristics of the participants who performed all tests.

Characteristics of Participants	Value
Sporting group (all, *n* = 113)	
Swimming, *n* (%)	33 (29.2)
Soccer, *n* (%)	30 (26.5)
Control, *n* (%)	30 (26.5)
Gymnastics, *n* (%)	6 (5.3)
Baseball, *n* (%)	3 (2.7)
American football, *n* (%)	3 (2.7)
Equestrian sports, *n* (%)	3 (2.7)
Lacrosse, *n* (%)	3 (2.7)
Tennis, *n* (%)	2 (1.8)
Age (year)	20.4 ± 1.8
Female sex, *n* (%)	43 (38.0)
Height (m)	1.68 ± 0.08
Weight (kg)	62.8 ± 9.3
Body mass index (kg/m^2^)	22.1 ± 2.0
Balance ability	
Modified index of postural stability	0.70 ± 0.25
One-leg standing time eyes closed (s)	53 (32, 60)
Muscle strength	
Knee extension strength (Nm/kg)	2.82 ± 0.52
Knee extension strength endurance (J)	1802 ± 502
Dorsiflexion strength (ankle joint) (kgf)	41.1 ± 10.4
Toe pinch force (kgf)	5.0 ± 1.5
Grip strength (kgf)	41.4 ± 9.8
Sensory tests	
Semmes–Weinstein monofilament test (g/mm^2^)	18.4 (7.2, 24.4)
Vibration perception test (sec)	16.4 ± 2.9
Body composition	
Skeletal muscle mass (kg)	29.3 ± 6.0
Body fat mass (kg)	11.2 ± 4.6

Data are in the form of mean ± standard deviation or median (25th, 75th).

**Table 5 healthcare-12-02240-t005:** A multiple regression analysis of mIPS-related variables with physical parameters was performed using a stepwise procedure.

R^2^ = 0.20	Estimate	95% Confidence Interval	*p* Value
Knee extension strength	0.12	0.04–0.21	0.006
Semmes–Weinstein monofilament test	−0.004	−0.008–−0.001	0.027
Height	−0.01	−0.01–−0.001	0.018
Dorsiflexion strength (ankle joint)	0.002	0.000–0.004	0.114

Covariates: all factors; mIPS, modified index of postural stability.

**Table 6 healthcare-12-02240-t006:** A multiple regression analysis of OLS-related variables with physical parameters was performed using a stepwise procedure.

R^2^ = 0.23	Estimate	95% Confidence Interval	*p* Value
Skeletal muscle mass	1.85	0.28–3.41	0.021
Height	−1.42	−2.14–−0.71	<0.001
Vibration perception test	0.95	−0.10–2.01	0.076
Knee extension strength	0.01	−0.001–0.02	0.065
Weight	−0.65	−1.41–0.12	0.099

Covariates: all factors; OLS, one-leg standing time with eyes closed.

## Data Availability

The authors will make the raw data supporting this article’s conclusions available upon request.

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
