# Peer review of "Balance Ability Characteristics and Related Factors in Athletes Across Different Sports: A Preliminary Study"

_healthcare, 2024, doi:10.3390/healthcare12222240_

Round 1

Reviewer 1 Report

Comments and Suggestions for Authors

- Revise the introduction by citing more previous studies. Currently, only a very limited number of studies are repeatedly referenced.

- I have concerns about whether the research objective and the analysis methods presented in this study are appropriate. Please review this section and either revise the research objective or adjust the analysis methods and present the results again.

- In the results, it is mentioned that participants could not complete the assessment due to the pandemic. Were they considered dropouts? Please provide a more detailed explanation of the procedures related to the pandemic.

- It seems that the variables used in Tables 2, 3, 5, and 6 are different, making it difficult to confirm the differences between sports categories.

- The analysis of balance and variables for each sport is not detailed enough, which raises the question of whether it is appropriate to describe the sports individually in the discussion.

- Proprioception is important for balance and athletes. I suggest that the discussion either address the lack of consideration for proprioception or incorporate it into the analysis.

Author Response

  1. Summary

  1. Questions for General

Evaluation Reviewer’s Evaluation Response and Revisions

Does the introduction provide sufficient background and include all relevant references?

Yes/Can be improved/Must be improved/Not applicable

Are all the cited references relevant to the research?  Must be improved

Is the research design appropriate?  Must be improved

Are the methods adequately described?  Can be improved

Are the results clearly presented?  Must be improved

Are the conclusions supported by the results?  Must be improved

  1. Point-by-point response to Comments and Suggestions for Authors

Reviewers' comments:

Reviewer #1:

The authors appreciate the care with which you reviewed our paper. Your comments were very helpful in improving this manuscript. If there are any further misunderstandings or our interpretations are inadequate, we will be happy to make further corrections or additions if you could point out any particular issues that remain. Your comments are highlighted in black bold. We are responding point-by-point as follows.

  1. Revise the introduction by citing more previous studies. Currently, only a very limited number of studies are repeatedly referenced.

Response 1

Thank you for the comment. In order to emphasize the necessity of balance ability in athletes, the importance of balance training was expanded in the introduction, and the argument that balance specificity for each sport is necessary for effective balance training was strengthened. In addition, the cited references were re-read carefully, and interpretations were revised. Necessary references were also added.

  1. I have concerns about whether the research objective and the analysis methods presented in this study are appropriate. Please review this section and either revise the research objective or adjust the analysis methods and present the results again.

Response 2

Thank you for pointing this out. The purpose of this study is described as "this study aimed to examine the balance characteristics (two-leg and one-leg types) for each competition using a unified quantitative evaluation index of balance ability including mIPS". Although there were some deficiencies in the variables (limitations due to the research environment) and variations in the number of participants for each competition in accordance with this "purpose", because this was an "exploratory study", the analysis method described was adopted (p.5, line 207-224).

However, this point was added because it is also a limitation of the study (see Research Design comment 2 in Reviewer #2).

------

Response 2 in Reviewer #2

Thank you for your comment. However, power for each level of sport is not easily calculated the relationships between levels, and because this is an exploratory research, a preliminary power analysis was not conducted. We have included it in the Discussion section as a future issue as follows:

“Second, there was an imbalance in the number of participants across sports groups. Sports with fewer than 10 participants were not included in the model analysis, so effects for these sports were not evaluated. Furthermore, because this was an exploratory study, the sample size was not designed based on statistical power, raising concerns that certain effects may not have been detected due to insufficient power. Increasing the sample size is recommended in future studies. “ (p.7 lines 460-465)

---------

  1. In the results, it is mentioned that participants could not complete the assessment due to the pandemic. Were they considered dropouts? Please provide a more detailed explanation of the procedures related to the pandemic.

Response 3

Thank you for your comment. As a typical research methodology, participants are not defined as dropping out unless they refuse to withdraw from the study. Originally, all participants were required to complete all assessment items. However, it is true that various circumstances arose due to the impact of the pandemic. We have added the following information about this situation to the result.

“The impact of the pandemic led to two main issues: (i) participants requested a shorter assessment time; (ii) the research team had to adjust the environment (limited the number of staff) to conduct the assessments. This resulted in an imbalance in the number of assessments for each sport.” (p.5 lines 235-239)

We have also added all missing data to the results:

“The following are the number of missing data points for each assessment item: knee extension strength (91), knee extension endurance (91), ankle joint dorsiflexion strength (97), toe pinch force (60), grip strength (90), SWMT (69), vibration perception test (60), and body composition (98). “ (p.5 lines 240-244).

  1. It seems that the variables used in Tables 2, 3, 5, and 6 are different, making it difficult to confirm the differences between sports categories.

Response 4

In the multiple regression models, variable selection was conducted using a stepwise method. The exploratory variables not selected are considered as having no significant contribution, so the effects of all explanatory variables can be evaluated. The following text has been added to the results section:

“The results of the multiple regression analysis are shown below. Explanatory variables that were not selected are considered to have had no significant effect on the outcome. “(lines 254-256)

  1. The analysis of balance and variables for each sport is not detailed enough, which raises the question of whether it is appropriate to describe the sports individually in the discussion.

Response 5

Thank you for your comment. The results of the multiple regression analysis showed that the coefficients of determination were R2 = 0.20-0.30, respectively, and the analytical accuracy was not high enough. In this regard, the following was added to the limiting factors:

“Fifth, the coefficient of determination in the multiple regression analysis was not high enough. Therefore, the accuracy of the analysis is limited, and further investigation based on this study’ finding is required.” (p7 lines 477-479)

  1. Proprioception is important for balance and athletes. I suggest that the discussion either address the lack of consideration for proprioception or incorporate it into the analysis.

Response 6

Thank you for your advice. As added in the results, only 113 subjects were able to complete all measurements, including the SWMT and vibration perception test, and we gave up because the N number was insufficient to perform an analysis for every sport. We have added a note about this in the limiting factors:

“Additionally, it was not possible to perform an analysis for each sport, nor to clarify the relationships between muscle strength, proprioception, and body composition, which are balance components specific to each sport.” (p.6 lines 455-458)

However, we have positioned these 113 subjects as "young people, including athletes," and as a basic resource, we have exploratory analyzed and considered factors related to mIPS and OLS:

“The participants in this factor analysis were young individuals, including athletes, and the results provide fundamental information for understanding the constituent factors of balance ability.” (p.5 lines 392-394)

Furthermore, we have newly taken up and added a note about training methods that focus on proprioception in the discussion:

“The second method is proprioception training. It has been reported that proprioception training for female volleyball players significantly improved their balance ability and competitive performance [40]. This suggests that proprioception has a causal relationship with balance ability and may be an effective training method for athletes.” (p.6 lines 437-441)

  1. Response to Comments on the Quality of English Language

Point 1: The quality of English does not limit my understanding of the research.

Response 1: The English text has been appropriately addressed and corrected.

We believe that we have addressed reviewers’ comments and hope that the revised manuscript is now acceptable for publication in Healthcare. Thank you for your generous consideration.

Sincerely,

Yasuhiro Suzuki

Department of Rehabilitation Medicine, University of Tsukuba Hospital

2-1-1 Amakubo, Tsukuba, Ibaraki 305-8576, Japan

Email: minaminagasaki2007@yahoo.co.jp

Phone: +81-29-853-3795

Fax: +81-29-853-7047

Reviewer 2 Report

Comments and Suggestions for Authors

Introduction and Background:
The introduction provides sufficient background on the importance of balance in athletic performance, referencing key studies that contextualize the study's goals. However, it could benefit from a more detailed exploration of why specific sports (such as judo and gymnastics) were chosen for comparison in balance capabilities. Including more recent references would enhance the relevance of the literature.

Research Design:
The study design is appropriate for the research question. The use of a cross-sectional approach and quantitative balance measures (mIPS and OLS) offers clear, interpretable data. However, the authors should clarify whether the sample size is adequate for detecting meaningful differences across sports, particularly given the low representation in some sports groups (e.g., only 2 participants in tennis). A discussion of power analysis would strengthen the justification for the sample size.

Methods:
The methods are generally well-described, but additional details on the selection and calibration of measurement tools, such as the mIPS and OLS, would enhance replicability. Specifically, explain why these two tests were chosen over other balance measures. The sensory and muscle strength tests, although appropriate, could benefit from further elaboration on how they were administered to ensure consistency across participants.

Results Presentation:
The results are clearly presented with appropriate use of tables and figures. However, the interpretation of the statistical findings could be expanded. For example, while the regression models are informative, the practical significance of the results (e.g., how much a change in knee extension strength affects balance capability) could be discussed more explicitly.

Conclusions and Interpretation:
The conclusions are supported by the results, but they would benefit from a deeper discussion of the limitations of the study. For instance, the lack of control for participants' skill levels within each sport is a significant limitation that may affect the generalizability of the findings. Additionally, while the paper highlights the role of height in balance, there is little discussion of how this could be accounted for in athletic training programs. Addressing these limitations in more detail would provide a more nuanced interpretation of the results.

Practical Applications:
The practical implications are briefly mentioned but could be expanded. The study would be more impactful if it offered specific recommendations for athletes and coaches regarding how to tailor balance training based on the findings. For instance, the paper could discuss whether athletes in different sports should prioritize two-leg or one-leg balance training.

Figures and Tables:
The figures and tables are generally clear, but some could benefit from additional labeling to clarify key comparisons. For instance, Table 2 could include footnotes explaining why certain sports groups (e.g., tennis) had fewer participants and how this might affect the interpretation of their results.

Author Response

  1. Summary

  1. Questions for General

Evaluation Reviewer’s Evaluation Response and Revisions

Does the introduction provide sufficient background and include all relevant references?

Yes/Can be improved/Must be improved/Not applicable

Are all the cited references relevant to the research?  Can be improved

Is the research design appropriate?  Can be improved

Are the methods adequately described?  Can be improved

Are the results clearly presented?  Can be improved

Are the conclusions supported by the results?  Can be improved

  1. Point-by-point response to Comments and Suggestions for Authors

Response to comments of Reviewer #2:

The authors appreciate the care with which you reviewed our paper. Your comments were very helpful in improving this manuscript. If there are any further misunderstandings or our interpretations are inadequate, we will be happy to make further corrections or additions if you could point out any particular issues that remain. Your comments are highlighted in black bold. We are responding point-by-point as follows.

Introduction and Background:

  1. The introduction provides sufficient background on the importance of balance in athletic performance, referencing key studies that contextualize the study's goals. However, it could benefit from a more detailed exploration of why specific sports (such as judo and gymnastics) were chosen for comparison in balance capabilities. Including more recent references would enhance the relevance of the literature.

Response 1

Thank you for your advice. This study was conducted with the aim of clarifying balance characteristics in various sports, but no specific sports were selected in a planned manner.

In other words, we recruited participants from various university sports clubs, and as a result, we positioned it as a "preliminary study" in which we selected the sports to analyze based on the attributes of the participants who applied. (However, only boat racing was pre-set.) This could have easily caused misunderstandings among readers, so we took the following measures.

Added to the title:

“Balance ability characteristics and related factors in athletes across different sports: a preliminary study”

Regarding two-leg type balance characteristics, we focus on the modified Index of Postural Stability (mIPS):

“Our previous research has shown that mIPS is associated with competitive performance in athletes, specifically boat racers [13, 14]. Therefore, mIPS can be considered an appro-priate index for assessing two-leg balance characteristic.” (p.2 lines 66-69)

Since this study is a preliminary one, the sports were not specified in advance, and the study was conducted on sports that were grouped together through participant recruitment. However, from the start, only boat racing, which has been shown to be associated with mIPS, was included as one of a target sport:

“However, due to the established association of mIPS with boat racing, it was included as one of the target sport.” (p.2 lines 87-89)

A total of 213 participants applied, resulting in 11 groups (control, judo, swimming, soccer, baseball, gymnastics, American football, equestrian sports, lacrosse, tennis, and boat race trainee). Seven groups (control, judo, swimming, soccer, baseball, gymnastics, and boat race trainee) had more than 10 participants.

“A total of 213 participants were included, devided into 11 groups: control, judo, swimming, soccer, baseball, gymnastics, American football, equestrian sports, lacrosse, tennis, and boat race trainee. Seven groups (control, judo, swimming, soccer, baseball, gymnastics, and boat race trainee) had more than 10 participants. These groups included 176 athlete participants (150 university students plus 26 boat race trainees; 107 men, 69 women; age, 20.2 ± 1.6 years; height, 1.69 ± 0.08 m; and weight, 68.0 ± 14.7 kg), and 37 non-athletes as control participants (university students) (Table 1).” (result: p.5 lines 223-229)

Research Design:

  1. The study design is appropriate for the research question. The use of a cross-sectional approach and quantitative balance measures (mIPS and OLS) offers clear, interpretable data. However, the authors should clarify whether the sample size is adequate for detecting meaningful differences across sports, particularly given the low representation in some sports groups (e.g., only 2 participants in tennis). A discussion of power analysis would strengthen the justification for the sample size.

Response 2

Thank you for your comment. However, power for each level of sport is not easily calculated the relationships between levels, and because this is an exploratory research, a preliminary power analysis was not conducted. We have included it in the Discussion section as a future issue as follows:

“Sports with fewer than 10 participants were not included in the model analysis, so effects for these sports were not evaluated. Furthermore, because this was an exploratory study, the sample size was not designed based on statistical power, raising concerns that certain effects may not have been detected due to insufficient power. Increasing the sample size is recommended in future studies. “ (p.7 lines 461-465)

Methods:

  1. The methods are generally well-described, but additional details on the selection and calibration of measurement tools, such as the mIPS and OLS, would enhance replicability. Specifically, explain why these two tests were chosen over other balance measures. The sensory and muscle strength tests, although appropriate, could benefit from further elaboration on how they were administered to ensure consistency across participants.

Response 3

Thank you for your comment. The reason for choosing mIPS has been explained above in the introduction.

“Regarding two-leg type balance characteristics, we focus on the modified Index of Postural Stability (mIPS) [12]. Our previous research has shown that mIPS is associated with competitive performance in athletes, specifically boat racers [13, 14]. Therefore, mIPS can be considered an appropriate index for assessing two-leg balance characteristic.” (p.2, lines 65-68)

The reason for choosing OFS has been explained above in the measurement:

“The advantage of OLS is that it can evaluate differences in competitive levels and can re-flect performance in specific sports, such as in relation to the level of proficiency in soccer [18, 19].” (p.3, lines 112-114)

Regarding “the sensory and muscle strength tests,” we have added the following explanation of how to conduct them:

“All measurements were performed in accordance with methods used in previous studies [20, 21] and were conducted consistently across all subjects.” (p.3, lines 124-125)

Results Presentation:

  1. The results are clearly presented with appropriate use of tables and figures. However, the interpretation of the statistical findings could be expanded. For example, while the regression models are informative, the practical significance of the results (e.g., how much a change in knee extension strength affects balance capability) could be discussed more explicitly.

Response 4

We appreciate your comment.

In this survey, knee extension strength, proprioception, height, and skeletal muscle mass were selected as factors related to balance ability. Based on these results, we have added a discussion of what kind of training will lead to improved balance ability, focusing on factors including muscle strength:

“The first approach is strength training. If strength training increases knee extension strength and muscle hypertrophy, it may improve the stability of the lower limb joints and neuromuscular coordination, indirectly contributing to enhanced balance ability. However, limited evidence on the direct effect of strength training on improving balance ability itself [9]. In other words, while muscle strength and muscle mass are necessary conditions for balance ability, strength training alone may have a limited impact.” (p.6 lines 431-437)

Conclusions and Interpretation:

  1. The conclusions are supported by the results, but they would benefit from a deeper discussion of the limitations of the study. For instance, the lack of control for participants' skill levels within each sport is a significant limitation that may affect the generalizability of the findings. Additionally, while the paper highlights the role of height in balance, there is little discussion of how this could be accounted for in athletic training programs. Addressing these limitations in more detail would provide a more nuanced interpretation of the results.

Response 5

Thank you for your suggestion. We have already mentioned in the limitations that we were not able to fully assess the athletes' competitive abilities, but we have added more explanation and references to strengthen the issue:

“Third, the competitive level of participating athletes in each sport was not clearly de-fined, so it could not be standardized. In some sports, differences in balance ability have been shown to depend on participant skill level, even within the same discipline [42-44]. “ (p.6-7 lines 466-469)

We have added a note in the discussion section regarding consideration of height when it comes to athletic training:

“Finally, it is essential to consider athlete’s height when designing training programs. Set-ting the difficulty level based on age standards for height can be beneficial, as height can influence changes in balance ability.” (p.6 lines 453-457)

Practical Applications:

  1. The practical implications are briefly mentioned but could be expanded. The study would be more impactful if it offered specific recommendations for athletes and coaches regarding how to tailor balance training based on the findings. For instance, the paper could discuss whether athletes in different sports should prioritize two-leg or one-leg balance training.

Response 6

Thank you for the comment. The relevant points have been added to the discussion:

“The third focus is balance training. This study revealed that balance types (two-leg and/or one-leg) vary by sport, making it important to design training programs specific to each type. For instance, it has been reported that balance training focused on one-leg training, tailored to the characteristics of the sport, may improve balance ability and technical skills in male soccer players[41]. If the balance characteristics determined in this study reflect the characteristics of the sport, mIPS and OLS are useful indicators for designing balance training.”( p.6 lines 441-447)

Figures and Tables:

  1. The figures and tables are generally clear, but some could benefit from additional labeling to clarify key comparisons. For instance, Table 2 could include footnotes explaining why certain sports groups (e.g., tennis) had fewer participants and how this might affect the interpretation of their results.

Response 7

Thank you for the comment. Regarding this point, please refer to Research Design “comment 2”.

  1. Response to Comments on the Quality of English Language

Point 1: The quality of English does not limit my understanding of the research.

Response 1: The English text has been appropriately addressed and corrected.

We believe that we have addressed reviewers’ comments and hope that the revised manuscript is now acceptable for publication in Healthcare. Thank you for your generous consideration.

Sincerely,

Yasuhiro Suzuki

Department of Rehabilitation Medicine, University of Tsukuba Hospital

2-1-1 Amakubo, Tsukuba, Ibaraki 305-8576, Japan

Email: minaminagasaki2007@yahoo.co.jp

Phone: +81-29-853-3795

Fax: +81-29-853-7047

Reviewer 3 Report

Comments and Suggestions for Authors

I am pleased to submit my review of the manuscript titled "Balance Capability Characteristics and Related Factors in Athletes Across Different Sports." This study presents a valuable exploration of the balance capabilities of athletes, examining the various factors that may influence these abilities across diverse sporting disciplines. Given the growing interest in the relationship between balance and athletic performance, this research contributes to a deeper understanding of how different sports may require distinct balance characteristics.

However, there are a number of issues that the authors need to take into consideration:

Abstract.

1. Consider rephrasing for clarity and conciseness. For example, instead of "examining the both-leg and single-leg balance characteristics of sports," it can be clearer to say "examining both leg and single-leg balance characteristics in athletes.“

2. If possible, include more specific information about the statistical methods used (e.g., significance levels, effect sizes) for a more comprehensive understanding.

3. Ensure consistency in terminology (e.g., "both-leg" vs. "bipedal") to avoid confusion.

4. Use transition words or phrases to enhance the flow of information between results (e.g., "In summary," "Additionally," or "Moreover") for smoother reading.

Materials and Methods

1. Ensure that the flow of sentences is logical. For instance, consider breaking down longer sentences for better readability.

2. Use consistent terminology (e.g., "non-athlete college students" instead of "non-athletes" later in the text).

3. When describing participant demographics, ensure that all relevant data points are presented in a consistent format (e.g., age, height, weight).

4. The exclusion criteria can be more clearly organized, perhaps using bullet points, for easier comprehension.

5. While the ethics statement is good, consider mentioning that the study adhered to ethical standards throughout the research process.

6. Change "was selected as the one-leg type balance characteristic" to "was selected to evaluate the single-leg balance characteristic" for clearer communication.

7. Use "two-leg balance characteristic" instead of "two-leg type balance characteristic" for conciseness.

8. Instead of "measurements included," use "was assessed through measurements of" to enhance professionalism.

9. Simplify "the sensation of vibration" to "vibration sensation" for brevity.

10. Replace "to avoid the ceiling effect" with "to minimize the ceiling effect" for more appropriate terminology.

11. Ensure smoother connections between the last sentences for easier reading, e.g., "Body composition was determined by measuring skeletal muscle mass and body fat mass."

References

- Make sure to follow the journal’s reference guidelines.

I appreciate the opportunity to engage with this important work and provide my insights.

Sincerely,

Comments on the Quality of English Language

I am pleased to submit my review of the manuscript titled "Balance Capability Characteristics and Related Factors in Athletes Across Different Sports." This study presents a valuable exploration of the balance capabilities of athletes, examining the various factors that may influence these abilities across diverse sporting disciplines. Given the growing interest in the relationship between balance and athletic performance, this research contributes to a deeper understanding of how different sports may require distinct balance characteristics.

However, there are a number of issues that the authors need to take into consideration:

Abstract.

1. Consider rephrasing for clarity and conciseness. For example, instead of "examining the both-leg and single-leg balance characteristics of sports," it can be clearer to say "examining both leg and single-leg balance characteristics in athletes.“

2. If possible, include more specific information about the statistical methods used (e.g., significance levels, effect sizes) for a more comprehensive understanding.

3. Ensure consistency in terminology (e.g., "both-leg" vs. "bipedal") to avoid confusion.

4. Use transition words or phrases to enhance the flow of information between results (e.g., "In summary," "Additionally," or "Moreover") for smoother reading.

Materials and Methods

1. Ensure that the flow of sentences is logical. For instance, consider breaking down longer sentences for better readability.

2. Use consistent terminology (e.g., "non-athlete college students" instead of "non-athletes" later in the text).

3. When describing participant demographics, ensure that all relevant data points are presented in a consistent format (e.g., age, height, weight).

4. The exclusion criteria can be more clearly organized, perhaps using bullet points, for easier comprehension.

5. While the ethics statement is good, consider mentioning that the study adhered to ethical standards throughout the research process.

6. Change "was selected as the one-leg type balance characteristic" to "was selected to evaluate the single-leg balance characteristic" for clearer communication.

7. Use "two-leg balance characteristic" instead of "two-leg type balance characteristic" for conciseness.

8. Instead of "measurements included," use "was assessed through measurements of" to enhance professionalism.

9. Simplify "the sensation of vibration" to "vibration sensation" for brevity.

10. Replace "to avoid the ceiling effect" with "to minimize the ceiling effect" for more appropriate terminology.

11. Ensure smoother connections between the last sentences for easier reading, e.g., "Body composition was determined by measuring skeletal muscle mass and body fat mass."

References

- Make sure to follow the journal’s reference guidelines.

I appreciate the opportunity to engage with this important work and provide my insights.

Sincerely,

Author Response

  1. Summary

  1. Questions for General

Evaluation Reviewer’s Evaluation Response and Revisions

Does the introduction provide sufficient background and include all relevant references?

Yes/Can be improved/Must be improved/Not applicable

Are all the cited references relevant to the research?  Can be improved

Is the research design appropriate?  Yes

Are the methods adequately described?  Can be improved

Are the results clearly presented?  Yes

Are the conclusions supported by the results?  Yes

  1. Point-by-point response to Comments and Suggestions for Authors

Response to comments of Reviewer #3:

Reviewer's opening comment

I am pleased to submit my review of the manuscript titled "Balance Capability Characteristics and Related Factors in Athletes Across Different Sports." This study presents a valuable exploration of the balance capabilities of athletes, examining the various factors that may influence these abilities across diverse sporting disciplines. Given the growing interest in the relationship between balance and athletic performance, this research contributes to a deeper understanding of how different sports may require distinct balance characteristics.

However, there are a number of issues that the authors need to take into consideration:

The authors appreciate the care with which you reviewed our paper. Your comments were very helpful in improving this manuscript. If there are any further misunderstandings or our interpretations are inadequate, we will be happy to make further corrections or additions if you could point out any particular issues that remain. Your comments are highlighted in black bold. We are responding point-by-point as follows.

Abstract.

  1. Consider rephrasing for clarity and conciseness. For example, instead of "examining the both-leg and single-leg balance characteristics of sports," it can be clearer to say "examining both leg and single-leg balance characteristics in athletes.“

Response 1

Thank you for the comment. I have corrected the relevant parts according to your suggestions (p.1 lines 15-16).

  1. If possible, include more specific information about the statistical methods used (e.g., significance levels, effect sizes) for a more comprehensive understanding.

Response 2

Thank you for the comment. Added estimate values in the result.

  1. Ensure consistency in terminology (e.g., "both-leg" vs. "bipedal") to avoid confusion.

Response 3

Thank you for the comment. It has been unified to "two-leg" and "one-leg".

  1. Use transition words or phrases to enhance the flow of information between results (e.g., "In summary," "Additionally," or "Moreover") for smoother reading.

Response 4

Thank you for the comment. I used "In summary," "Additionally," to make the flow between sentences smoother.

Materials and Methods

  1. Ensure that the flow of sentences is logical. For instance, consider breaking down longer sentences for better readability.

Response 1

Thank you for the comment. As per your suggestions, some sentences have been split and revised.

  1. Use consistent terminology (e.g., "non-athlete college students" instead of "non-athletes" later in the text).

Response 2

Thank you for the comment. The relevant sections have been unified to "non-athletes."

  1. When describing participant demographics, ensure that all relevant data points are presented in a consistent format (e.g., age, height, weight).

Response 3

Thank you for the comment. We have strengthened the relevant parts as you pointed out. We have also organized the information:

“A total of 213 participants were included, devided into 11 groups: control, judo, swimming, soccer, baseball, gymnastics, American football, equestrian sports, lacrosse, tennis, and boat race trainee. Seven groups (control, judo, swimming, soccer, baseball, gymnastics, and boat race trainee) had more than 10 participants. These groups included 176 athlete participants (150 university students plus 26 boat race trainees; 107 men, 69 women; age, 20.2 ± 1.6 years; height, 1.69 ± 0.08 m; and weight, 68.0 ± 14.7 kg), and 37 non-athletes as control participants (university students) (Table 1).” (p.5 lines 223-229)

  1. The exclusion criteria can be more clearly organized, perhaps using bullet points, for easier comprehension.

Response4 

Thank you for the comment. We have addressed the relevant points in bullet points:

“(i) spontaneous nystagmus; (ii) visual impairment or limb movement disorders affecting daily living; (iii) inability to maintain normal standing posture; (â…³) dizziness or vertigo; (V) history of equilibrium sensory disorder; (â…µ) history of falls; (â…¶) and lack of independence in walking and daily living.” (p.2-3 lines 96-100).

  1. While the ethics statement is good, consider mentioning that the study adhered to ethical standards throughout the research process.

Response5 

Thank you for the comment. I have corrected the relevant parts according to your suggestions:

“Additionally, this study adhered to ethical standards throughout the research process.” (p.3 lines 105-106)

  1. Change "was selected as the one-leg type balance characteristic" to "was selected to evaluate the single-leg balance characteristic" for clearer communication.

Response6 

Thank you for the comment. I have corrected the relevant parts according to your suggestions. (p.3 lines 110)

  1. Use "two-leg balance characteristic" instead of "two-leg type balance characteristic" for conciseness.

Response7 

Thank you for the comment. I have corrected the relevant parts according to your suggestions.

  1. Instead of "measurements included," use "was assessed through measurements of" to enhance professionalism.

Response8 

Thank you for the comment. I have corrected the relevant parts according to your suggestions (p.3 lines 119-120) (p.3 lines 121-122).

  1. Simplify "the sensation of vibration" to "vibration sensation" for brevity.

Response9 

Thank you for the comment. I have corrected the relevant parts according to your suggestions (p.3 lines 122).

  1. Replace "to avoid the ceiling effect" with "to minimize the ceiling effect" for more appropriate terminology.

Response10 

Thank you for the comment. I have corrected the relevant parts according to your suggestions (p.3 lines 112-113).

  1. Ensure smoother connections between the last sentences for easier reading, e.g., "Body composition was determined by measuring skeletal muscle mass and body fat mass."

Response11 

Thank you for the comment. The problem has been corrected as follows:

“Body composition was assessed by measuring skeletal muscle mass and body fat mass and was measured using bioelectrical impedance analysis (InBody 720; Biospace, Tokyo, Japan).” (p.5 lines 200-202).

References

  1. Make sure to follow the journal’s reference guidelines.

Response 1

Thank you for the comment. I have corrected everything as suggested.

  1. Response to Comments on the Quality of English Language

Point 1: The quality of English does not limit my understanding of the research.

Response 1: The English text has been appropriately addressed and corrected.

We believe that we have addressed reviewers’ comments and hope that the revised manuscript is now acceptable for publication in Healthcare. Thank you for your generous consideration.

Sincerely,

Yasuhiro Suzuki

Department of Rehabilitation Medicine, University of Tsukuba Hospital

2-1-1 Amakubo, Tsukuba, Ibaraki 305-8576, Japan

Email: minaminagasaki2007@yahoo.co.jp

Phone: +81-29-853-3795

Fax: +81-29-853-7047

Reviewer 4 Report

Comments and Suggestions for Authors

The number of sports registered and their difference in sample size, including some without female profiles, makes the results weaken the results.

It is suggested to reduce the sample to sports containing a sample close to the control sample and with both male and female gender profiles.  

Furthermore the focus of the discussion would be more interesting to propose the methodology as a tool for use in measuring sport performance in a control fashion in the sports where it is suggested that uni- and bilateral balance is so important.  

It is described but with an obvious conclusion. 

Author Response

  1. Summary

  1. Questions for General

Evaluation Reviewer’s Evaluation Response and Revisions

Does the introduction provide sufficient background and include all relevant references?

Yes/Can be improved/Must be improved/Not applicable

Are all the cited references relevant to the research?  Yes

Is the research design appropriate?  Yes

Are the methods adequately described?  Yes

Are the results clearly presented?  Yes

Are the conclusions supported by the results?  Yes

  1. Point-by-point response to Comments and Suggestions for Authors

Response to comments of Reviewer #4:

The authors appreciate the care with which you reviewed our paper. Your comments were very helpful in improving this manuscript. If there are any further misunderstandings or our interpretations are inadequate, we will be happy to make further corrections or additions if you could point out any particular issues that remain. Your comments are highlighted in black bold. We are responding point-by-point as follows.

  1. The number of sports registered and their difference in sample size, including some without female profiles, makes the results weaken the results.

It is suggested to reduce the sample to sports containing a sample close to the control sample and with both male and female gender profiles.

Response 1

Thank you for the comment. The fact that gender differences were not uniform among the athletic groups participating in this study was a major limiting factor, and has been added to the limitations section:

“Fourth, the gender distribution in each sports group was unbalanced. Females may have lower physical strength and athletic performance than males, potentially affecting the results. Since this was an exploratory study, adjustments were made for female participants; however, for conclusive findings, the gender of participants should be standardized. “(p.7 lines 472-476)

  1. Furthermore the focus of the discussion would be more interesting to propose the methodology as a tool for use in measuring sport performance in a control fashion in the sports where it is suggested that uni- and bilateral balance is so important.

It is described but with an obvious conclusion.

Response 2

Thank you for the comment. As an extension of these results, we are considering incorporating them into our training programs. We have added this point to the discussion. (p.6 lines 428-450)

  1. Response to Comments on the Quality of English Language

Point 1: The quality of English does not limit my understanding of the research.

Response 1: The English text has been appropriately addressed and corrected.

We believe that we have addressed reviewers’ comments and hope that the revised manuscript is now acceptable for publication in Healthcare. Thank you for your generous consideration.

Sincerely,

Yasuhiro Suzuki

Department of Rehabilitation Medicine, University of Tsukuba Hospital

2-1-1 Amakubo, Tsukuba, Ibaraki 305-8576, Japan

Email: minaminagasaki2007@yahoo.co.jp

Phone: +81-29-853-3795

Fax: +81-29-853-7047

Round 2

Reviewer 1 Report

Comments and Suggestions for Authors

The authors have acknowledged the limitations pointed out by the reviewer and incorporated them into the manuscript. Therefore, the response to the comments is sufficient. I agree with the publication of the manuscript.